# Case Report of an Interprofessional Intervention to Improve Quality of Life for a Fluid-Limited Patient

**DOI:** 10.3390/pharmacy10010018

**Published:** 2022-01-21

**Authors:** Jennifer L. Cox, Maree Donna Simpson

**Affiliations:** School of Dentistry and Medical Sciences, Charles Sturt University, Orange, NSW 2800, Australia; masimpson@csu.edu.au

**Keywords:** aged care, pharmacist medication review, interprofessional, quality of life

## Abstract

This was a case of an 81-year-old female, an amputee, who presented at hospital with a fractured neck of femur after a fall in the nursing home. The patient was being treated for several complex chronic conditions for which 30 regular medicines were prescribed and 100 tablets were being taken per day. The patient was fluid limited to 1500 mL per day but the need to swallow such a high number of tablets meant that there was no fluid allowance available for any other drinks, not even a cup of tea. In the nursing home, the patient had multiple prescribers, not all from the one surgery. The pharmacist conducted a multifaceted review of the patient’s medication and lifestyle factors. Working collaboratively with the wider health care team, the intervention was able to reduce the number of medications and improve the patient’s quality of life through improving the effectiveness of other lifestyle factors. This case not only showcases pharmacist interventions but also the synergistic benefit of interprofessional working with patients with chronic and complex conditions. This is arguably more critical in rural or remote areas where there is commonly a paucity of most health practitioners, health assistants and technicians.

## 1. Introduction

Inappropriate prescribing of medication in elderly patients is an important and on-going public health issue [1,2,3]. Research has shown that over half of the residents in Australian aged care facilities are prescribed at least one potentially inappropriate medicine [4]. Multiple co-morbidities and an increased number of prescribers contribute to the high prevalence of polypharmacy in aged care [5] and may impose burdens on the affected residents and on staff to administer those medicines. Many residents in residential institutional care with multiple chronic conditions are prescribed and use increasingly complex medication regimens. Simplifying medication regimen complexity by using sustained-release products or dose forms that are alternatives to oral forms may offer many benefits including improved health-related quality of life.

The case study presented here aligns with a shift in focus from treating patient illness to managing consumer health and well-being [6] and demonstrates the potential benefits of embedding pharmacists into collaborative health care teams described in the Pharmacists in 2023 report [7]. These benefits include improved decision making for the safe and appropriate use of medicines, reduced polypharmacy for aged care patients, more proactive care at the point of initiating and modifying medicine treatment, and improved links between prescriber and dispenser [7].

## 2. Case Presentation

The patient was an 81-year-old female who had a low body weight and was encountered in hospital with a fractured neck of femur resulting from a fall in the nursing home. The fracture was encouraged to knit rather than being treated surgically with a hip replacement. Several years ago, following a right below-knee amputation (from a traumatic accident), the patient had been given an older-style wooden prosthetic leg. The leg, however, was poorly fitted and 5–6 socks had to be put over the stump to manage the resulting discomfort. The patient walked using the stump and a hand-held wooden walking stick (right hand). This was an unstable situation for walking and staying upright as only one hand was free to open doors and pick up things.

The patient was fluid limited as a “solution” to nocturia and incontinence. A disposable bed pad was used in her bed and incontinence pads were worn if the patient needed to leave the home. Other than that, there are by design many toilets in the nursing home, so a toilet was usually close by, but that sudden incentive to move swiftly was effectively a falls risk waiting to happen. The patient was able to dress herself (carefully) and observe daily hygiene such as toileting, but assistance was needed for showering.

The patient’s history described the following conditions/issues:Chronic lower back pain,Restless legs syndrome (RLS),Phantom limb pain,Irritable bowel syndrome,Diverticulitis,High blood pressure,Depression,Glaucoma in both eyes,Gastro-oesophageal reflux disease (GORD),Osteoporosis,Benign paroxysmal positional vertigo (BPPV), andInsomnia.

This patient was observed to be very pragmatic and did not readily complain, however stated a strong desire for a “proper prosthesis” and the ability to have a cup of tea.

The patient was fluid limited to 1500 mL per day, however had a daily regimen of 30 regular medicines and 100 tablets which needed to be taken one at a time, with ample water. Minor swallowing issues meant that a gulp of water was required for each tablet, effectively meaning that this patient had no fluids left as a drink—so no cup of tea nor juice with breakfast.

The patient was, however, being treated for more than this and seemed unaware of her significant falls risk arising from many contributing factors including physiological aging; untreated glaucoma; significant polypharmacy with drowsiness, confusion, or dizziness as side effects, poorly supportive and fitting wooden prosthesis, use of a wooden cane in the right hand, pain medicines, urinary incontinence, and rush to get to a toilet.

The patient was fiercely independent and made the best of her situation; however, social exclusion was evident—no family lived close by and no visitors were recorded for the duration of the hospitalisation. Social exclusion is a multidimensional concept that identifies socio-economic variables that contribute to inequality and injustice [8]. Contributing variables include poverty, poor health, and limited social networks. In health, access to medicines and care may vary, sometimes significantly, in public hospitals from private hospitals.

In this case study, the patient was hospitalised geographically distant from her community and did not receive any friends or family who might have provided some resources and engagement to cope with a longer stay in hospital. In addition, other contributing variables such as modest means and poor health also applied to her and impacted her perceptions of support and assistance.

The pharmacy who provided this case study supplied the private hospital and the nursing home with medicines, so institutionally rather than just this patient. Of concern, the nursing home had multiple prescribers, not all from the one practice. There was an agreement between the General Practitioners (GPs) that whichever GP was at the nursing home on the day would order medicines for every patient needing them even if they were not his/her patient before entering the nursing home.

### Pharmacist Intervention

This was a complex and chronic situation requiring a multistage whole-person management approach. The first step was to obtain a complete picture of medications. Table 1 shows the medications that the patient was taking at the time of hospitalisation.

The review of medications identified areas where multiple medications, often with opposing actions, had been prescribed. The actions/suggestions resulting from the initial review resulted in five fewer medicines for the patient (Table 2). A review of the patient’s prosthesis at a public hospital was also recommended.

Stage 2 involved seeking a case conference with all or the primary prescriber and addressing other conditions such as the glaucoma, enhancing the effectiveness of medications and reducing falls risk (Table 3).

Additional suggestions:Refer to pelvic floor physiotherapist for assessment of incontinence and possible pelvic floor exercisesSeek incontinence support (Continence Foundation of Australia) to institute CAPS (Continence Aids Payment Scheme) and provide incontinence garments

## 3. Outcome

Polypharmacy, most commonly in patients over the age of 65, taking more than 5 medicines, has been linked not to better health outcomes but rather to negative consequences such as falls, frailty, and mortality [9]. Medication reviews are often requested for older persons with multimorbidity and polypharmacy to reduce medication-related problems. In the Netherlands, Verdoom and colleagues [10] established the utility of clinical medication review and of incorporation of patient goals and preferences. They reported improved life experience, well-being and quality of life. This is consistent with the experience of our patient when not only her medication was addressed but also her preferences.

As a result of the intervention, this patient now has a modern, well- fitting prosthesis, and has a wheeled walker for additional support. Of importance, the medication regimen now has a reduced number of doses within medicines as well as a reduced number of medicines and the patient can afford to have approximately 200 mL of her choice of fluid. A swallow of water is approximately 15 mL and an adult female mouthful is approximately 50 mL. After review of her incontinence (urologist, physiotherapist) which is still to come, the patient may not need to be fluid restricted (post-pelvic floor and incontinence garments, plus well-fitting prosthesis) and then would be able to consume fluid to 2 litres, i.e., an extra 500 mL (a cup and a small glass of fluids). So, the patient can have a small cup of tea now and may be able to have a small hot milk at night and peaches with juice for dessert. Inter-professional collaboration and patient management have made this possible. For example, a speech pathologist and dietician were engaged to address the swallowing advice and dietary assistance to reduce constipation through dietary interventions, in addition to a stool softener/laxative product. A referral to an optometrist addressed eyesight issues that may have contributed to falls and falls risk plus previously untreated glaucoma. Further, two specialist physiotherapists provided care for walking with greater safety, incontinence and BPPV. This enhanced the pharmacy interventions and our patient’s quality of life [11].

## 4. Conclusions

Here, we report a case of polypharmacy and other lifestyle factors within which the comprehensive pharmacist review generated an inter-professional case discussion and a multifaceted intervention that not only optimised the effectiveness of the patient’s medication but alleviated or ameliorated other factors such as falls risk, dry eyes, and cracked skin. Medicine simplification protocols may reduce the burden of medication administration for aged care providers and provide enhanced quality of life for patients.

This case study highlights the pivotal role that a pharmacist can play in initiating holistic management of complex patients and demonstrates the positive impacts of integration of health care teams, especially for those individuals with co-morbidities and complex conditions.

## Figures and Tables

**Table 1 pharmacy-10-00018-t001:** Medications grouped according to symptom.

Symptom	Medication at Time of Hospitalisation
Constipation	Coloxyl 120 (stool softener); Macrogol (osmotic laxative); Bisacodyl (stimulant laxative)
	Targin (oxycodone/naloxone) (morning); Endone (oxycodone) (evening)
Insomnia	Clonazepam (benzodiazepine)
Neuropathic pain	Lyrica (pregabalin)
Hypertension	Irbesartan—angiotensin II receptor blocker
Dry skin (prone to cracking)	Two different steroids (one cream, one ointment); 10% urea cream
Chronic cough	Bromhexine—Mucolytic; Pholcodine-Opium alkaloids and derivatives
Overactive bladder/Urinary incontinence	Oxybutynin (urinary antispasmodic); Mirabegron (urinary antispasmodic)
RLS	Pramipexole (dopaminergic anti-Parkinson’s disease agent)
Depression	Venlafaxine—selective serotonin and norepinephrine reuptake inhibitor
	Potassium oral supplement
GORD	Pantoprazole (proton pump inhibitor) at highest dose, twice a day

**Table 2 pharmacy-10-00018-t002:** Stage 1 recommendations.

Action/Suggestion	Rationale
Changed irbesartan to amlodipine	To deal with chronic cough
Stopped two cough mixtures	Different modes of action which act at cross purposes
Start a long-term clonazepam decreasing dose regimen to cessation	
Cease potassium supplement	Pathology tests showed normal potassium levels
Cease all medicines for constipation	
Start regular Coloxyl and senna	Stool softener plus stimulant laxative
Suggested oxybutynin be reviewed	Patient stated that it made her feel dizzy; recognised adverse effects of dry mouth, dry eyes quite challenging for a fluid-restricted patient
Recommended cessation mirabegron	Most common adverse effects include hypertension, urinary tract infection, dry mouth and constipation

**Table 3 pharmacy-10-00018-t003:** Stage 2 review and recommendations.

Action/Suggestion	Rationale
Review and assess glaucoma (optometrist or ophthalmologist referral needed)	Rural location of patient means ophthalmologist referrals can take many months, so optometrist referral prioritised
Use Systane Ultra for dry eyes	Management of dry eyes until ophthalmologist review
Propose change from venlafaxine (depression) to duloxetine (depression and pain)	Manage two conditions with 1 medication, reduce number of medicines/doses
Reduce pantoprazole to 20 mg at night and consider Gaviscon liquid for breakthrough heartburn	Use of PPI may contribute to increased risk of any fractures and she has osteoporosis.
Consider small hot milk drink with dinner for sleep	
Cease pramipexole and replace with levodopa/benserazide nocte only	Severity of RLS was lower during the time of falling asleep and during the night and satisfaction and disease burden was also higher
Phantom limb pain—move pregabalin to bedtime	To lower falls risk
Severe dry skin—Shower chair, use soap-free wash and also moisture shield after patting dry. Review two steroids that were being used regularly. Consider moving from moderate (triamcinolone) and potent (mometasone) to hydrocortisone and keeping stronger for flare-ups	
BPPV—cease prochlorperazine, review with vestibular physiotherapist for procedure and exercises	Reduce severity or resolve condition
Change pain management to oxycodone-with-naloxone-controlled-release (CR) tablets (Targin) lower strength in the morning and higher strength at bedtime	

## Data Availability

The datasets generated during and/or analyzed during the current study can be find in the main text.

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
