# Peer review of "Case Report of an Interprofessional Intervention to Improve Quality of Life for a Fluid-Limited Patient"

_pharmacy, 2022, doi:10.3390/pharmacy10010018_

Round 1

Reviewer 1 Report

The revised article must be improved in all its points to be considered a research report. Theoretical sections are very scarce, practically without bibliographic citations, there is no section on methodology and results, nor for discussion. The article needs a considerable revision and improvement.

Author Response

Comment: The revised article must be improved in all its points to be considered a research report. Theoretical sections are very scarce, practically without bibliographic citations, there is no section on methodology and results, nor for discussion. The article needs a considerable revision and improvement.

Response: Thankyou for your feedback. This article was specifically submitted as a case study, rather than a research report and has been written as such. 

Reviewer 2 Report

I read with interest the paper entitled “I’d just about kill for a cuppa”: Case report of an interprofessional intervention to improve quality of life for a fluid-limited patient”. It deals with the very actual item of deprescribing in older adults. Medication overload or problematic polypharmacy is a very important problem causing harm, especially in older adults. Taking multiple medications rises the risk of potentially inappropriate medications, and residents in long-term care are frequently prescribed several medications at once. It is necessary to intervene to address this problem and physicians and/or pharmacists should perform regular medication reviews, even if this process is complicated and time-consuming. Because of prescription review, medications may be reduced, changed, or stopped altogether. Different methods have been suggested in order to reach such an important target, however, it has not been defined the best one yet. Different nations worldwide have different healthcare systems, therefore the method for reviewing polypharmacy would be decided locally based on the available resources. This case report demonstrates such a problem. Typing “deprescribing” in the PubMed search string selecting the case reports section, gives only 66 items. The case report reported in the paper is very interesting and it worthy to be published, especially because it stimulates the intervention of different health care professionals including optometrist and physiotherapist. I have only minor point to discuss:

  • Line 52 “see for example”, I cannot understand the reason for these words
  • Line 73 I would add BPPV after “Benign paroxysmal positional vertigo”
  • Lines 149-156 I think that the paragraph would not be necessary in a case report paper
  • Lines 158-165 Informed Consent Statement should contain the fact that patients agreed in publishing her own data
  • Some words appear to be very informal (colloquial language), therefore non-native English speaking readers (as I am) could find some difficulties in understanding the sentence.

Author Response

Point 1:

Line 52 “see for example”, I cannot understand the reason for these words

These words have been removed

Point 2: Line 73 I would add BPPV after “Benign paroxysmal positional vertigo”

Acronym added as suggested

Point 3: Lines 149-156 I think that the paragraph would not be necessary in a case report paper

Lines 149-156 removed

Point 4: Lines 158-165 Informed Consent Statement should contain the fact that patients agreed in publishing her own data

There was no ethics application submitted as this case was an in-practice intervention as part of normal practice (for which permission was
received from the pharmacist to make a de-identified case). Accordingly, the case was de-identified to protect ethical issues such as
privacy/confidentiality of practitioners and patient. As such, it was agreed by the editor that Informed consent was not required.

The “Informed Consent Statement” section now reads “Patient consent was waived due to this case being an in-practice intervention as part of normal practice (for which permission was received from the pharmacist to make a de-identified case). Accordingly, the case was de-identified to protect ethical issues such as privacy/confidentiality of practitioners and patient.

Point 5: Some words appear to be very informal (colloquial language), therefore non-native English speaking readers (as I am) could find some difficulties in understanding the sentence.

Agreed. Document reviewed and colloquial words amended to more formal words where appropriate.

Reviewer 3 Report

Thank you for allowing me to review this article. This case study describes an all too often and unfortunate example of the care that elderly patients receive in substandard nursing home facilities.  Thank you for taking your time to document it here for the readers of Healthcare. 

I provide below my comments and recommendations for improving or strengthening the rigor of this case study.

Background:

Throughout the case presentation, the tense should be presented in the past form. “She can dress herself….”  The authors could say: “Patient was able to dress and observe daily hygiene such as toileting, with a raised toilet…”

Possibly rephrase throughout as the “patient” instead of the “lady”, as it is more objective.

This sentence is not grammatically correct: This lady was noticed to be very pragmatic and did not readily complain, however if asked for “if onlies – as in-if only I could…”, she will admit she would like a “proper prosthesis” AND a cup of tea – she really misses that.

Suggest: The patient was observed to be pragmatic and did not readily complain; however, she stated “if onlies [sic]”– (meaning, “if only I could…”), she would like two things: a “proper prosthesis” and a cup of tea. She really missed having a cup of tea.

Starting p.2, line 51: a “solution” has a tone that could be made more objective.

No need for words in ALL CAPS (e.g., RIGHT, AND).

Unless words are a direct quote from the patient or other (i.e., clinician), do not use quotation marks for emphasis. This implies subjectivity that reduces the rigor of your writing. The rule applies for exclamation marks, which do not belong in research writing unless they are attached to a direct quotation. This may bias your reader.

Make sure to explain all abbreviations before using them (e.g., GP, GORD, PPI, RLS).

Explain why lack of visitors is mentioned and how this is related to the patient’s case: reasons given could be 1) her welfare, 2) her mental capacity, 3) health literacy, or 4) patient advocacy.

Explain why public versus private institution is mentioned, and how this is related to the patient’s case (see above for suggested line of thought). Explain medical indigence as appropriate.

Under pharmacist interventions, I suggest the authors cluster her medications by symptomology or by disease state. As is, there is no logic to seeing the volume of medications other than reviewing a long list. It would be effective for clinicians to see the repetition of medications for the same or similar condition, then your interventions would follow a more analytical pathway.

Unfortunately, the word “cuppa” does not translate well to international readers, so I would reserve this as a colloquialism and use another phrase such as “chosen beverage” (without quotes) or the patient may have a beverage of her choice, including the hot tea she expressed formerly.

In the conclusion, you mention interprofessional case discussions; however, this is a pharmacy intervention. You must discuss the other healthcare professions involved in order for this to be truly interprofessional.

I recommend you state the implications of polypharmacy comparing this case to other published cases, and describe the change in the quality of life for the patient.     

IRB: Do you have IRB approval or was this exempt? You need a statement here.

Informed Consent: You need a statement here. (I assume consent was waived.)

Author Response

Thankyou so much for your constructive feedback. Please see the attachment

Reviewer 4 Report

The authors report on a very interesting case report. However, the manuscript is written in a rather dramatic way, than scientific. I recommend a drastic change of writing style and resubmission.  

Author Response

Comment:

The authors report on a very interesting case report. However, the manuscript is written in a rather dramatic way, than scientific. I recommend a drastic change of writing style and resubmission.

Response:

A number of amendments have been made to the manuscript in line with feedback from reviewers 2 & 3. The authors believe these amendments have improved the writing style of the manuscript

Round 2

Reviewer 1 Report

The authors have made the proposed changes.

Author Response

No changes required

Reviewer 3 Report

The term health "cover" is not universally understood as a noun. Health "coverage" might be better. I first read this as a typo.  

Thank you for your edits. The paper is greatly improved as a result of your work and additions of peer reviewed research. Nice job. I have one minor suggestions and one comment. I applaud this work on many levels as a clinician and as a researcher. 

Table 1 greatly adds to the article and is an interesting component. Excellent format. 

Suggested revisions to Lines 153- : For example, to a speech pathologist and dietician, were engaged to address..

So this would read: For example, a speech pathologist and dietitian were engaged to address...

Author Response

Thankyou for your feedback. Line 153 amended as suggested.

Reviewer 4 Report

Authors made significant changes in their work, but this is still not enough. Authors should understand this decision as support for the development of their scientific writing and invest more time in the adaptation of this work. I recommend the implementation of all given advice and resubmission of work. Furthermore, I recommend that consultation with the Editor regarding Patient consent. 

Title: *“I’d just about kill for a Cuppa”:* not acceptable from my point of view, as part of a title. 
Abstract: "..81-year-old woman..." patient or female would sound maybe better"..an amputee.." not acceptable - refer more as with amputation or similar. 
"...she had multiple prescribers..." not she but the patient..

1. Introduction - well done, scientific writing great. 
2. Case Presentation
"...The patient is an 81-year-old woman.." female or patient.
"...She was not given a hip replacement..." The patient did not... not she. "She" must be in the whole text omitted, otherwise from my point of view not acceptable.
"...She was fluid limited, not because of heart failure or similar, but rather as a “solution” to nocturia and incontinence...." -> not because of heart failure or similar,  to be removed as does not have any impact. 
"...She could dress herself (carefully) but needed assistance showering and drying her hair and body...." replacement of description of nursing activities could be better replaced with *personal hygiene* instead of listing it. 
"This patient was observed to be very pragmatic and did not readily complain, however, she stated “if onlies [sic]” – (meaning “if only I could…”), she would like two things: a “proper prosthesis” and a cup of tea. She really missed having a cup of tea. -> *She* is not acceptable in any part of this sentence. Reporting on the meaning of what is the patient state is better than citing her own words. 
Restless legs exist only as Restless legs syndrome
IBS (Irritable bowel syndrome) - shortcut not needed, report only full sentence.
Heartburn (Gastro-oesophageal Reflux Disease, GORD) - report only with medical terms
Vertigo (Benign paroxysmal positional vertigo, BPPV) - report on one of both, shortcut only needed when later mentioned. 
in the whole text using word swallow should be reconsidered with the more medical terms: "...which she needed to swallow one at a time.." -> "which she needed to *take* one at a time"
"1.5L (1500mL)" report on one or another unit. 
Line 95-97 - reformulate, can every patient with emergency condition (heart attack / stroke?!) decide into which hospital to go or they would be taken to the nearest hospital with for example Stroke unit or Cardiac intervention center? Delete or reformulate. In an emergency, the status of patient insurance should not be of importance, furthermore, it could be that patient is due to severity of urgent condition not able to share his/her wish.
Line 100: link in the text is not acceptable, this has to be referenced. 
Line 107-112-> not the best-organized health system, are the authors really sure that they would like to make this kind of statement for an Australian health care system?
Table 1 needs to be revised according to the usual scientific writing rules. 
"Pantoprazole (proton pump Inhibitor) at highest strength and high dose, twice a day." what is the difference between the highest strength and dose? It would be better to report on the dose, which can be different depending on region. 
 Restless Legs Syndrome (RLS) - do shortcut only or explanation only. 
" To deal with chronic cough" this is not an acceptable medical term. reformulate.
"Pathology normal" how can pathology be normal?
"Cease all medicines for constipation, review diet and refer for di-etitian consult" and next line state use of new laxatives.
Line 127. "heartburn" not to be used 
Line 127: BPPV - shortcuts need to be explained at the last line of the table.
Line 127: Targin - use always generic medication with products.
3. Outcome: Incontinence is no indication for serious fluid intake limitation (if otherwise, please provide references). Why is the patient still fluid restricted after the whole revision? 

Informed Consent Statement: "Anonymisation means that neither the patient nor anyone else could identify the patient." Patient or relatives could identify this patient due to a very detailed description of patient needs and disability. The explanation is not enough, refer the link: https://www.mdpi.com/journal/pharmacy/instructions#ethics. Furthermore, other case reports did report on the Ethical committee: https://www.mdpi.com/2226-4787/8/4/212/htm
